# Evidence for low density holes in Jupiter's ionosphere

Masafumi Imai [1], Ivana Kolmašová [2,3], William S. Kurth [1], Ondřej Santolík [2,3], George B. Hospodarsky[1], Donald A. Gurnett[1], Shannon T. Brown[4], Scott J. Bolton[5], John E.P. Connerney[6,7] & Steven M. Levin[4]

Intense electromagnetic impulses induced by Jupiter's lightning have been recognised to produce both low-frequency dispersed whistler emissions and non-dispersed radio pulses. Here we report the discovery of electromagnetic pulses associated with Jovian lightning. Detected by the Juno Waves instrument during its polar perijove passes, the dispersed millisecond pulses called Jupiter dispersed pulses (JDPs) provide evidence of low density holes in Jupiter's ionosphere. 445 of these JDP emissions have been observed in snapshots of electric field waveforms. Assuming that the maximum delay occurs in the vicinity of the free space ordinary mode cutoff frequency, we estimate the characteristic plasma densities (5.1 to $250\,cm^{-3}$) and lengths ($0.6\,km$ to $1.3\times10^{5}\,km$) of plasma irregularities along the line of propagation from lightning to Juno. These irregularities show a direct link to low plasma density holes with $\leq250\,cm^{-3}$ in the nightside ionosphere.

[1] Department of Physics and Astronomy, University of Iowa, Iowa City, IA 52242, USA. [2] Department of Space Physics, Institute of Atmospheric Physics, The Czech Academy of Sciences, 117 20 Prague, Czechia. [3] Faculty of Mathematics and Physics, Charles University, 110 00 Prague, Czechia. [4] Jet Propulsion Laboratory, California Institute of Technology, Pasadena, CA 91125, USA. [5] Space Science and Engineering Division, Southwest Research Institute, San Antonio, TX 78238, USA. [6] Space Research Corporation, Annapolis, MD 21403, USA. [7] NASA Goddard Space Flight Center, Greenbelt, MD 20771, USA. Correspondence and requests for materials should be addressed to M.I. (email: masafumi-imai@uiowa.edu)

Jovian whistlers detected by the Voyager 1 plasma wave instrument[1] provided incontrovertible evidence of lightning at Jupiter[2]. The interpretation of the Jovian whistlers as lightning was based on the comparison to terrestrial lightning-induced whistlers[3], and the independent observations of optical lightning flashes by Voyager 1[4]. A lightning-induced non-dispersed electromagnetic pulse was observed at Jupiter by the Galileo Probe in a magnetic field waveform in the frequency range from 10 Hz to 100 kHz[5]. Since their discovery intense electromagnetic impulses induced by Jupiter's lightning have been recognised to produce both low-frequency dispersed whistler emissions[2,6,7] and non-dispersed radio pulses[5,8]. Jovian whistlers appear between a few tens of Hz and 20 kHz[6,7], propagating below the local electron cyclotron frequency $f_{ce}$ or the local electron plasma frequency $f_{pe}$, whichever is lower according to the definition of the whistler mode[9]. Recently, Juno has also detected lightning-induced radio pulses called sferics at 600 MHz and 1.26 GHz[8]. However, Voyager radio observations in a frequency range from 20 kHz to 41 MHz reported no detections of radio pulses in the Jovian inner magnetosphere[10]. Non-detections of this kind were interpreted as strong radio absorption in the Jovian ionosphere[10].

Here, we present dispersed millisecond pulses with a lower frequency cutoff between 20 and 150 kHz recorded by the Juno radio and plasma wave (Waves) instrument[11] during eight perijove passes (closest approaches to Jupiter) from perijove 1 (PJ1) on 27 August 2016, through PJ9 on 24 October 2017[12–15]. In accounting for the dispersion curves of these pulses, we use a free space ordinary (O) mode straight-line propagation model (Methods and Supplementary Fig. 1), which assumes the presence of plasma density irregularities along Juno's line of sight. Because the occurrence positions of these pulses are collocated with those of Jovian lightning-induced whistlers[7] and 600-MHz sferics[8] independently detected by Juno, these irregularities correspond to an ionospheric plasma density less than 250 cm$^{-3}$. On the basis of the theory of lightning-induced microsecond trans-ionospheric pulse pairs on Earth[16–18], we suggest that the upper limit of the vertical height between the thunderstorm and the reflection layer in the Jovian atmosphere might be less than 500 km.

## Results

**Observations of Jupiter dispersed pulses.** We have carried out a survey of the Juno Waves burst mode data from the Low Frequency Receiver High (LFR-Hi) channel in the form of frequency-time spectrograms below 150 kHz (Methods) on PJ1 and PJ3 to PJ9. We found 445 instances of unusual discrete, dispersed pulses within 210 snapshots out of the total 58,542 available snapshots acquired below 5.5 Jovian radii ($R_J$, 1 $R_J$ = 71,492 km). All of the pulses were detected while Juno was at altitudes between 9790 km and 316,000 km above the 1 bar level. Figure 1 illustrates various types of spectral structures showing dispersion such as a pair of pulses (Fig. 1a), a train of four discrete pulses (Fig. 1b), a long dispersed pulse (Fig. 1c), and a short dispersed pulse (Fig. 1d). To the best of our knowledge, none of the previous literature reports such pulses, so we call them Jupiter dispersed pulses (JDPs), hereafter.

In the framework of cold plasma theory[9], there are two observations that lead to the conclusion that JDPs propagate in the free left-hand ordinary (L-O) mode. The first is that, while they generally occur below $f_{ce}$ (98% of the time), they can be found above $f_{ce}$ (about 2% of the time). The small percentage of cases $> f_{ce}$ is at least partly due to the upper frequency limit of the Waves LFR-Hi band, 150 kHz. Emission above $f_{ce}$ eliminates the whistler mode and leaves either the Z or L-O mode. Another observation is that JDPs can be found above the maximum Z-

mode frequency, the upper hybrid frequency $f_{uh} = \sqrt{f_{pe}^2 + f_{ce}^2}$; for example, the maximum frequency of the JDP in Fig. 1c is at least 150 kHz, well above both $f_{ce} = 126$ kHz and $f_{uh} = 132$ kHz (assuming $f_{pe} = 40$ kHz as the highest intensity of the smooth emission below the JDP).

To further understand the nature of JDPs, we have manually digitised spectral shapes of all 445 detections. In Fig. 2a, a histogram of JDP detections (left axis) and cumulative probability (right axis) is plotted as a function of duration with a 0.2 ms step. The peak occurs at the bin whose centre is 0.1 ms, and 95% cumulative probability is achieved within 3.2 ms. Figure 2b depicts another histogram for the inter-pulse spacing. It utilises only snapshots containing two or more pulses allowing inter-pulse spacing. Hence, only 53% of the detections (236 counts) are included. In this distribution, there are two peaks (Supplementary Fig. 2) appearing at bins with central values of 0.7 ms and 3.3 ms, and 95% cumulative probability (50% probability from the complete set of all detected pulses) occurs within 7.4 ms.

**Modelling JDP spectral shapes.** Illustrations of the fit results using the O mode straight-line propagation model (Methods and Supplementary Fig. 1) are shown as orange curves in Fig. 1, clearly capturing the spectral morphology of JDPs. Additional support for the fit results comes from a positive correlation between $f_{pe0}$ estimated from the model and $f_{cutoff}$ measured from the spectrograms (Supplementary Fig. 3). Figure 2c shows the results for length $D$ versus plasma density $N_{e0}$ of plasma density irregularities along a line of sight from a source location to Juno in a log–log scatter plot. The distributions show a systematic tendency of increases in $N_{e0}$ as $D$ decreases due to the strong negative correlation of determinations of both parameters in the model (Supplementary Fig. 4). Nevertheless, these distributions provide possible solutions to characterise the JDP spectral shapes (Supplementary Fig. 5). Specifically, the length $D$ of the irregularity structures ranges from 0.60 km with a corresponding $N_{e0}$ of 110 cm$^{-3}$ through $1.3 \times 10^5$ km with a corresponding $N_{e0}$ of 8.0 cm$^{-3}$. In Fig. 2d, we re-organise the distributions of $D$ versus Juno altitude. It is clear that the estimated $D$ is mostly smaller than the altitude, which is consistent with a hypothesis that the JDP radio sources are located in Jupiter's atmosphere in the same hemisphere as seen from Juno.

**Comparison of whistlers, sferics, and JDPs.** We compare the source locations of Jovian whistlers and sferics with the sources of the JDPs. The Juno Waves instrument detected 1627 whistlers between 27 August 2016, and 1 September 2017 (PJs 1 through 8)[7]. Assuming that whistlers propagate parallel to the magnetic field (so-called ducted whistlers)[3], the footprints of these whistlers can be estimated by mapping from Juno's position along the modelled JRM09 magnetic field lines[19] onto the Jovian atmosphere at 300 km altitude (the bottom edge of the Jovian ionosphere measured by Voyager 2[20] and Galileo[21]) above the 1-bar level. Similarly, the Juno Microwave Radiometer (MWR) instrument[22] originally captured 377 lightning sferics in a narrowband channel at 600 MHz within 100-ms integration intervals[8]. More recently, the sferic catalogue has been revised with 383 detections related to MWR antenna calibration. Figure 3a shows the Jovian whistler source locations using the ducting assumption as the orange plus marks in Jovian System III coordinates. As the MWR boresight pinpoints the lightning source within the beam projected onto the 1 bar surface, the yellow stars are estimates of the 600-MHz sferic source locations. The JDP vertically projected locations are shown as blue circles in Fig. 3a. Also, the latitudinal histogram of the detection rate with 5° bins

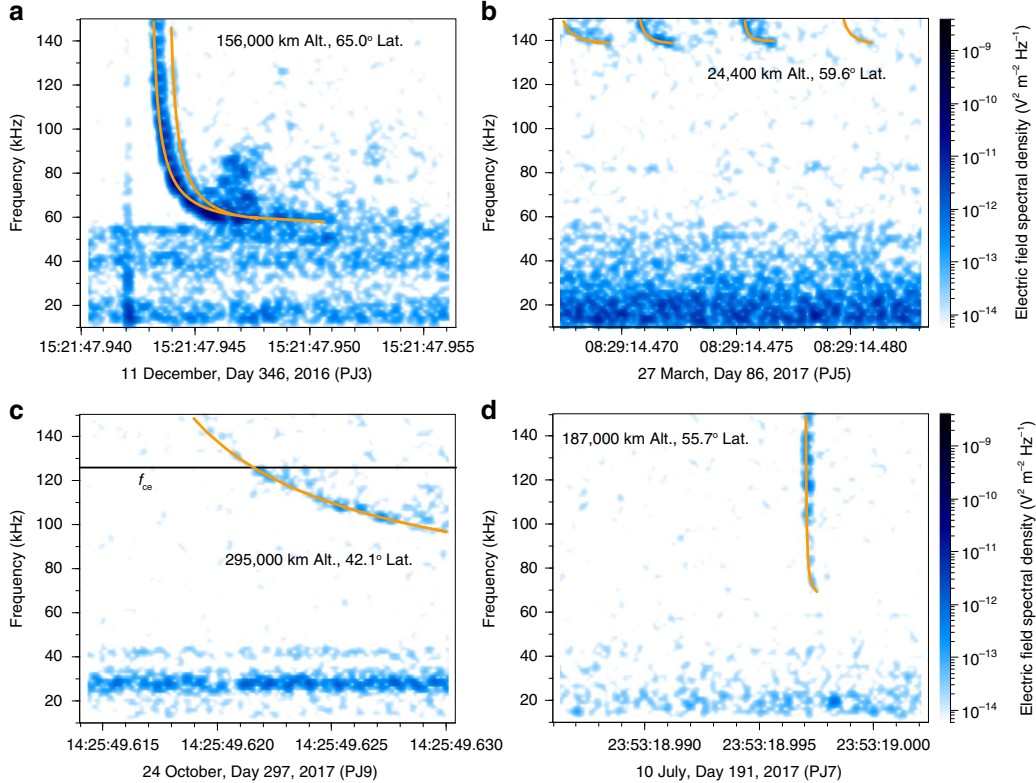

**Fig. 1** Four examples of Jupiter dispersed pulses (JDPs). These spectrograms were converted from 16.384-ms waveform snapshots. The orange curves are fitted with three free parameters $D$, $f_{pe}$, and $C$ via the O mode propagation model (Methods and Supplementary Fig. 1). There are a pair of JDPs (**a**) with a pulse-to-pulse interval of 0.7 ms, four individual JDPs (**b**) with a high cutoff frequency ($f_{pe} = 138$ kHz), one JDP (**c**) with a large dispersion ($D/c = 54$ ms), and one JDP (**d**) with a short dispersion ($D/c = 0.18$ ms). The black line in **c** shows the local electron cyclotron frequency $f_{ce}$ computed from the Juno's on-board magnetometer[29], while $f_{ce}$ in the other examples is well above the upper frequency range of the LFR-Hi burst mode, 150 kHz

for JDPs and whistlers and with 0.5° bins for sferics, is included on Fig. 3b using corresponding colours. The JDP, whistler and sferic distributions are similar in that both occurrence rates are higher in the northern hemisphere than in the southern hemisphere. The longitude range of 0° to 210° in the southern hemisphere shows no JDPs, contrary to the whistler and sferic distributions. Even if we consider the limited LFR-Hi observation coverage, this trend remains (Supplementary Fig. 6). Hence, the lack of JDPs could be due to a dense ionosphere between the JDP radio source and Juno.

## Discussion

In addition to two concurrent JDP-sferic events (Supplementary Fig. 7), the JDP source locations are similar to those of lightning-induced whistlers and 600-MHz sferics, suggesting JDPs are related to lightning. Given that JDPs are driven by Jupiter's lightning, it is challenging to address the question of how the low-frequency radio signals propagate from the atmosphere to the magnetosphere through the dense ionosphere. At Jupiter, the peak of the ionospheric plasma frequency is about 1–5 MHz[20,21], which is much higher than the frequency of the emissions studied in this paper. There are two possible interpretations for the JDP detections. The first is that the lightning-induced waves start out in the L–O mode, couple into the Z mode in the ionosphere, and then couple back into the L–O mode in the topside ionosphere. However, it is likely unfavourable due to a very inefficient conversion of the two mode-couplings. Another interpretation is that the sferic signal escapes through low electron plasma density holes (<250 cm$^{-3}$) in the ionospheric layer before reaching Juno. If we follow this interpretation, our estimated $N_{e0}$ is a model of

the upper bound of ionospheric plasma density irregularities below Juno.

Figure 4 shows $N_{e0}$ of the JDP detections plotted as a function of Jovian latitude and longitude. Recall that, while the 600-MHz sferics propagate freely through the dense ionosphere, the JDPs can be seen only when there is a low density path through the ionosphere that allows these pulses to reach Juno's position. While there is some uncertainty in the exact JDP radio source locations, we assume that the JDP radio sources can be vertically projected from Juno onto the Jovian atmosphere. The low plasma density holes in the ionosphere tend to appear more in the northern hemisphere than in the southern hemisphere. No JDPs are observed around the Jovicentric equator because the local plasma frequency of the Jovian ionosphere tends to exceed the upper recordable frequency of 150 kHz at latitudes from −10° to 25° and the JDP L-O mode waves cannot propagate in this region.

Our current knowledge of Jovian ionospheric profiles relies heavily on a radio occultation technique[23], as it integrates the plasma density in the transverse direction toward Earth on Jupiter's day side. It is also known that the peak of the ionosphere estimated by Voyager 2[20] and Galileo[21] varies significantly in altitude and plasma density. Another indirect observational study of the Jovian ionosphere was carried out via dispersion analysis using Juno's detections of lightning-induced whistlers. In this analysis, for some cases, it was necessary to reduce the iono-spheric plasma density model by 10 to 30%[7,24]. In other words, the Jovian ionosphere changes dynamically. Juno's first nine orbits used in this study were concentrated near the terminator (Supplementary Fig. 8). It is possible that the actual radio loca-tions of JDPs observed by Juno and postulated ionospheric holes are on the night side, where a different recombination process

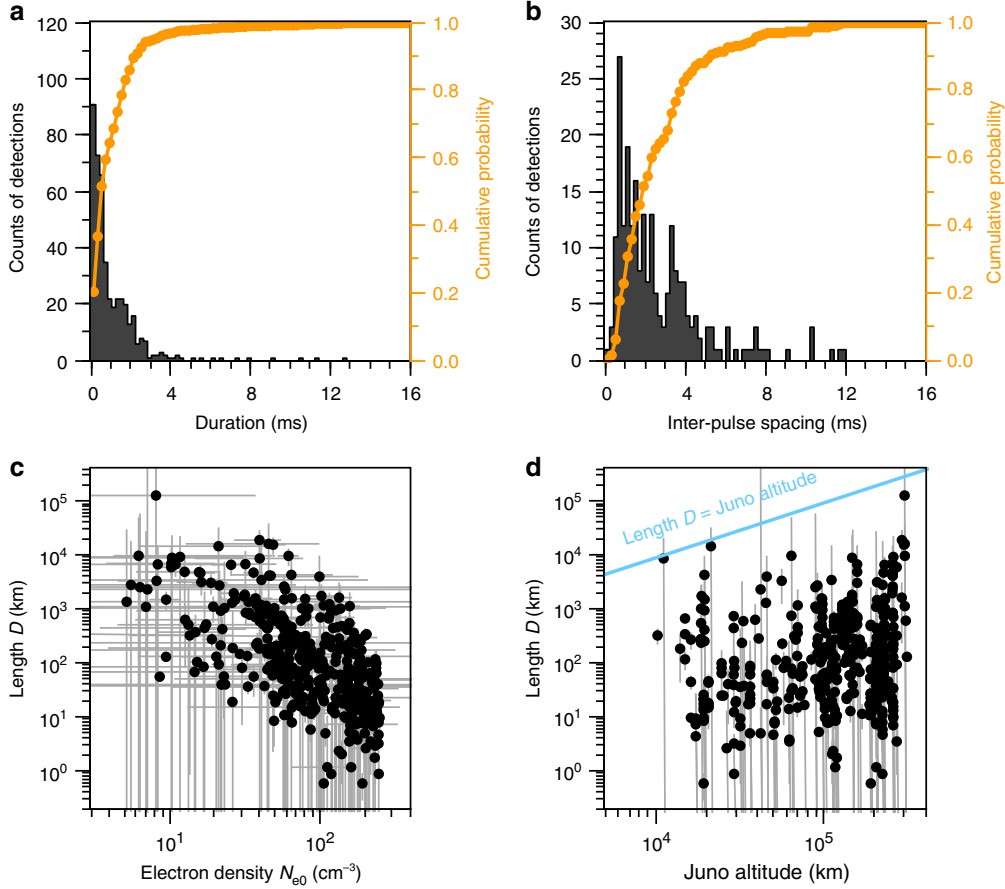

**Fig. 2** JDP properties based on Juno's observations and the propagation model. Distributions of JDP detections are organised as a function of **a** duration and **b** inter-pulse spacing using the left-hand axes. Each bin of both histograms is 0.2 ms. Using the right-hand axes, the orange lines are displayed for the corresponding cumulative probability. Note that the maximum of 16 ms for both duration and inter-pulse spacing is due to the length of the Waves LFR-Hi waveform snapshot. Density irregularities are modelled by a step function with height $N_{e0}$ and length $D$ using the O mode straight-line propagation model. The estimates of length $D$ are plotted as a function of **c** $N_{e0}$ and **d** Juno altitude using logarithmic scales. The grey error bar corresponds to one standard deviation (68% confidence interval) of $N_{e0}$ and $D$. The median values of the one standard deviation from the model fittings are 27.5% ± 2.5% for $D$ and 2.5% ± 2.5% for $N_{e0}$ (Supplementary Fig. 9)

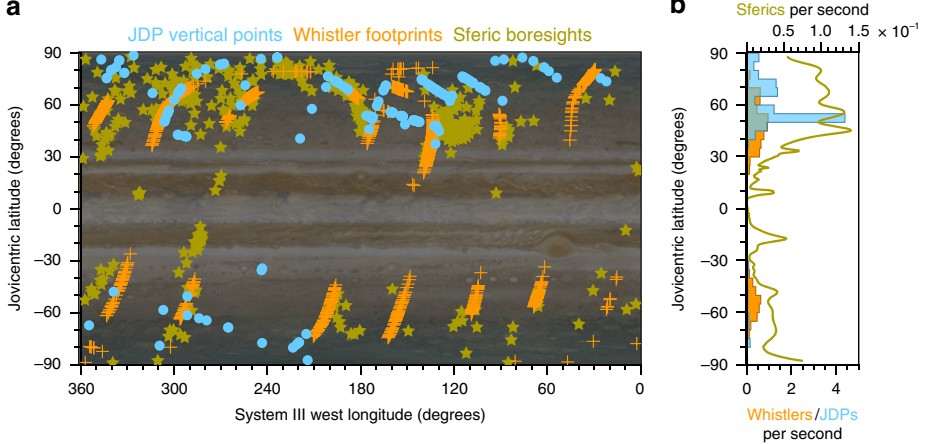

**Fig. 3** Jovian global map of JDPs, lightning-induced whistlers and sferics. **a** The blue circles depict the vertical projection of Juno when JDPs were captured. The orange plus marks indicate the whistler footprints that were mapped along the JRM09 magnetic field lines[19] onto the Jovian atmosphere at altitude of 300 km above the 1-bar level. The yellow stars are the MWR boresights at the 1-bar level for the detections of sferics at 600 MHz. These data were taken from the Juno Waves whistler catalogue[3] and the Juno MWR sferic catalogue[5] from perijoves 1 through 8. Note that the region between 200° and 280° where JDPs present were sampled from the Juno perijove 9 orbit where sferic and whistler observations have not been completed. The Jovian image was provided by NASA/JPL-Caltech/SSI/SwRI/MSSS/ASI/INAF/JIRAM/Björn Jónsson (http://www.planetary.org/multimedia/space-images/jupiter/merged-cassini-and-juno.html). **b** Their latitudinal profiles of detection rate are shown using corresponding colours. In addition, the comparison of JDPs with the previous optical detections of lightning is shown in Supplementary Fig. 10

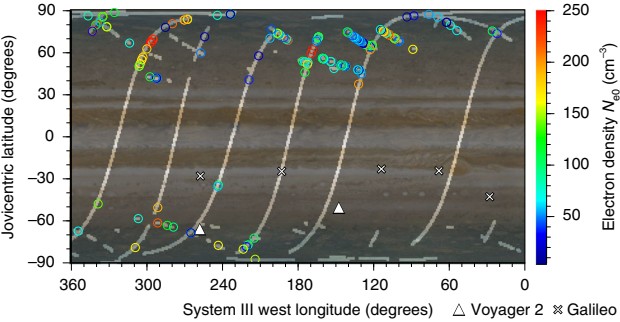

**Fig. 4** Estimated electron density $N_{eO}$ of the JDP locations using the propagation model. This profile is the upper bound ionospheric density because the JDP L–O mode waves should transect Jupiter's ionosphere and plasma irregularities from the lightning stroke to Juno. The transparent regions correspond to vertical projections of Juno when the Waves LFR-Hi observations were made. The observational locations of radio occultation measurements are depicted as triangles for Voyager 2[20] and crosses for Galileo[21]. The Jovian image was provided by NASA/JPL-Caltech/SSI/SwRI/MSSS/ASI/INAF/JIRAM/Björn Jónsson (http://www.planetary.org/multimedia/space-images/jupiter/merged-cassini-and-juno.html)

from the day side is anticipated[23]. In addition, because the natural presence of ionospheric holes has been widely recognised in Venus[25], Mars[26], and Saturn[27,28], the similarity to Jupiter can be inferred.

The frequency-time spectral shapes of JDPs (see especially Fig. 1a) are reminiscent of those of the trans-ionospheric pulse pairs (TIPPs) detected by Earth-orbiting satellites[16–18]. TIPPs are pairs of dispersed pulses induced by short duration intracloud lightning discharges and observed in a frequency band from 25 to 80 MHz with a duration of a few microseconds and a pulse-to-pulse interval of tens of microseconds. An interpretation for the appearance of a pair of pulses is that the radio signal branches into a direct pulse as the first pulse and an indirect pulse via ground reflection as the second pulse[17]. The inter-pulse interval is the differential travel time equal to the combination of twice the lightning source distance from the ground and the angular separation between the first and the second pulses, divided by the speed of light. Unlike Earth, Jupiter has no ground but has deeper layers, including the hypothetical possibility of a reflection layer well below the water clouds at the 5 bar level where we place the anticipated location of the source lightning discharges. Applying this interpretation to the JDP inter-pulse intervals of 0.7 ms and 3.3 ms, the apparent distances are respectively 100 and 500 km, which include distance-dependent angular separation but give an upper limit of the vertical distance between the lightning thunderstorm and the reflection layer. Another possible scenario for the observed inter-pulse intervals can be linked to individual strokes with typical repetition periods at 0.7 and 3.3 ms. The synoptic observations of JDPs from Juno, in combination with analyses of Jovian whistlers and sferics, will improve our understanding of the physical process of lightning discharges in Jupiter's atmosphere where in-situ measurements are limited.

## Methods

**Juno waves data used in this study**. One of the instruments onboard Juno is a radio and plasma wave instrument (Waves)[11], designed to monitor the electric fields of waves from 50 Hz to 41 MHz with an electric dipole antenna and the magnetic fields of waves from 50 Hz to 20 kHz with a magnetic search coil sensor using three on-board receivers. One of the receivers is the Low Frequency Receiver (LFR), recording three different components: one low-frequency (LFR-Lo) electric field component and one magnetic field component both from 50 Hz to 20 kHz and one high-frequency (LFR-Hi) electric field component from 10 kHz to 150 kHz. The LFR-Hi burst mode used in this study obtains 6144 points with a temporal resolution of 85 microseconds in a 16.384-ms waveform snapshot once

per second. Using a 256-point fast Fourier transform (FFT) on the ground, spectral data can be obtained covering 10 to 150 kHz with a spectral resolution of 1.5 kHz. In addition, we obtain the local electron cyclotron frequency converted from the measurements of the magnetic field recorded by the Juno's magnetometer[29].

**O mode straight-line propagation model**. In accounting for the frequency-dependent dispersion curve of JDPs, we use an O mode straight-line propagation model in which the maximum delay occurs in the vicinity of the O mode cutoff frequency, the electron plasma frequency $f_{pe0}$ in kHz, in a plasma density irregularity along Juno's line of sight. Supplementary Fig. 1 shows the modelled geometry of Juno, a radio source, and a plasma density irregularity in Jupiter's ionosphere or inner magnetosphere, where an enhanced electron plasma density $N_{e0} = (f_{pe0}/8.98$ kHz cm$^{3/2})^2$ in cm$^{-3}$ with a length of $D$ km is located along a straight line from the source through Juno at a distance of $L$ km. The observed time $t(f)$ for JDP can be expressed via a group delay[30] as

$$
\begin{aligned}
t(f) &= \int_0^L \frac{ds}{v_g} + t_0 = \int_0^{D_l} \frac{ds}{c} + \int_{D_l}^{D_h} \frac{ds}{v_g} + \int_{D_h}^L \frac{ds}{c} + t_0 \\
&= \frac{D_h - D_l}{c\sqrt{1 - \frac{f_{pe0}^2}{f^2}}} + \frac{L - D_h + D_l}{c} + t_0 \\
&= \frac{D}{c\sqrt{1 - \frac{\left(8.98\,\text{kHz cm}^{3/2}\right)^2 \times N_{e0}}{f^2}}} + C
\end{aligned} \tag{1}
$$

where $D_h$ and $D_l$ are the straight-line distances to the upper and lower boundaries, respectively, of the plasma density irregularity, $v_g$ is the group velocity of the O mode, $c$ is the speed of light, $t_0$ is the wave generation time, $f$ is the observed frequency, and $C$ is $(L - D)/c + t_0$. By using a non-linear least-squares fit of the digitised $t(f)$ points, three free parameters, $N_{e0}$, $D$, and $C$ have been estimated for each JDP. It is important to note that $C$ is just an offset due to a lack of determination of the exact source location because we cannot uniquely determine $L$ and $t_0$. But $N_{e0}$ and $D$ provide beneficial information on the structure of the plasma density irregularities. Eight examples of simulated dispersed pulses are displayed in Supplementary Fig. 5.

## Data availability

The Juno data used in this study are publicly accessible through the Planetary Data System (https://pds.nasa.gov). The catalogues that support the findings of this study are available from the corresponding author upon reasonable request.

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

## Acknowledgements

The authors are grateful to all members of the Juno mission team who make the Juno mission possible, especially the engineers and staff of the Juno Waves instrument. The research at the University of Iowa was supported by NASA through Contract 699041X with the Southwest Research Institute. The work of I.K. and O.S. were supported by MSM100421701 and LTAUSA17070 grants, and by the Praemium Academiae award from the Czech Academy of Sciences.

## Author contributions

M.I. and I.K. independently searched JDPs in all of the Waves waveform snapshots, which were incorporated into a common catalogue of these pulses. They also produced the Waves whistler catalogue. W.S.K. is the lead-investigator of the Waves instrument, helping to understand the nature of radio and plasma waves at Jupiter recorded by the instrument. O.S. and G.B.H. helped to organise and understand the morphology of JDPs. D.A.G. helped to suggest the physical explanation of JDPs. S.T.B. provided the 600-MHz sferic catalogue using Juno MWR data. S.J.B. is the principal investigator of the Juno mission. J.E.P.C. provided the Juno magnetic field data. S.M.L. is the lead-investigator of the MWR instrument. Incorporating feedback from all of the co-authors, M.I. wrote this manuscript.

## Additional information

**Competing interests:** The authors declare no competing interests.

