## [Peer Review File · Nature Communications]

Reviewers' comments:

Reviewer #1 (Remarks to the Author):

The dispersed pulses in Jovian atmosphere should prove to be of big interest to the scientific community, especially if as the authors suggest they are propagating through the ionospheric density holes.

In short, the paper is well-written and structured and I recommend publication after minor fixes.

Below follows a list of comments and suggestions, in case the attached PDF fails.

L51: A short introduction is necessary for most readers, describing why JDPs are interesting and what they can reveal

L56: "Jovian radii" - add $1R_J=...$ for convenience?

L66: perhaps a short recap of the modes?

L71: $f_{pe} = 40$ kHz - what is the basis of this assumption?

L73: phenomenology = nature?

L78: what about the other 47%?

(do they have more than 16 ms between? why not include them too?)

L80: which original set?

L86-88: from the figure it seems that there are actually two distributions, this is very interesting and should be discussed

L97: this is very specific, why 300km?

L108-111: this is a bias then, can it be corrected for? perhaps by filtering the longitudes with high auroral emissions (and then showing in an extra figure)?

L112: bad start of a sentence, consider rephrasing,

e.g., "the locations of the JDP source are.... suggesting...." or similar

L128-132: Question: Is it a "trivial" knowledge that the lightnings occur all the time, but the pulses are seen only when an ionospheric "hole" is present allowing the wave to pass? If this is the case, I think it would be useful to highlight this somewhere here - using JDPs as hole detectors should prove useful.

Figure 2: consider recoloring the cum. prob. axes to orange for consistency

Figure 2: considering that there are no counts above 13 ms, the histograms could be "zoomed" a bit to make the peaks more distinguishable - unless 16 ms is somehow important?

L182-183: errorbars in the figure to match these would make interpretation much easier, even if it's just one generic "legend" of uncertainty (i.e., for one point)

Figure 3: the axis on the right plot is a bit too far from the labeled y-axis, consider adding tick values here as well

Supplementary Figure 1: with this orientation of the figure, all of the y-axes appear upside down, which is a bit inconvenient to read - consider rotating the figure clockwise, or rotating the y-axes labels

Supplementary Figure 2: the blue bars seems superfluous (and confusing), if the sferic events are detected all the time during these intervals, why not just indicate this?

Reviewer #2 (Remarks to the Author):

Review for "Evidence for low density holes in Jupiter's ionosphere" by Imai et al.

This paper presents the first detection of a new form of radio emission, Jupiter dispersed pulses. The authors hypothesise that these radio pulses are linked to jovian lightning bursts due to the coincidence of events with sferics and Whistler waves.

The paper is interesting - new radio emission forms are quite exciting! - and worthy of publication after the following points are addressed:

Line 78: Figure 2b contains only 53% of the detections as it focuses on repeating JDP pulses with a repetition time less than 16 ms. What about the remaining 47% of the data? Are those events single pulses? What is the importance of this inter-pulse spacing? Does it strengthen the case for lightning bursts driving the emission?

Line 106 - 111: What about the region between 200 to 280 degrees where JDPs are present in the southern hemisphere, but neither sferics nor Whistlers are observed? There are additional locations where there are unique events of each type. It's not obvious that the correlation between JDPs and sferics is as strong as suggested by the authors.

Line 130 - 132: Is the lack of JDPs observed around the jovicentric equator an observational artefact due to the upper recordable frequency of the instrument? This upper limit has already been mentioned in line 68 as the reason that only 2% of cases are seen with frequencies above fce. Clarification is needed.

Supplementary figure 3 and associated discussion in lines 141 - 143: Due to Juno's orbit, measurements of JDPs and sferics is biased towards the terminator. The discussion should be tempered with respect to the local time coverage or that bias should be more explicitly stated in text. The phrasing "it is possible that the radio locations of JDPs observed by Juno and postulated

ionospheric holes are both on the night side where a different recombination process from the day side is anticipated” is unclear.

Reviewer #3 (Remarks to the Author):

Comments on “Evidence for low density holes in Jupiter’s ionosphere” by Imai et al

This reviewer found that the paper could be interesting to multidisciplinary readers but the work lacks strong and definitive evidence and analysis to support the conclusions.

Major comments:

1. The authors assumed the sources for the JDPs were produced by lightning below Juno, which might be correct but needs more definitive data to unambiguously support their claim. In figure 3 the authors compared the Juno positions when the JDPs were detected with the whistler and MWR (500 MHz) source positions and claimed that the JDPs were detected in the same general regions and inferred them as related to lightning. It is not clear from the main text and the supplementary material if the three types of observation were coordinated in time and space. Coordinated or not, the majority of the JDPs appear to be detected along different Juno orbital trajectories as compared to the other two observations. If they were coordinated, one should expect a MWR detection for each of the JDPs. The authors provided two events in Supplementary figure 2 that appear to be detected at the same time (within 100 ms) and position by MWR. However, 2 out of 445 appears to be a very small fraction and the authors need to rule out the chance of random coincidence.

2. The dispersion model (assuming it good enough for the ionosphere) used by the authors in Method is more sensitive to D than to Ne0 to fit the observed dispersion curve, therefore, one would expect D to have a smaller scattering range than Ne0. However, figure 2c shows the opposite with D scattering in ~4 orders of magnitude and Ne0 in ~2 orders of magnitude. In addition, the paper lacks the necessary error analysis for D, Ne0 and C, making the critical results of Ne0 less confident. If a Taylor expansion is used to approximate the authors’ model, one can see that the dispersion is proportion to the product of D and Ne0 to the first order, and therefore will naturally leads to the general feature of figure 2c (regardless of the scattering ranges), i.e., a smaller D will require a larger Ne0. So the authors’ statement in line 86 “The distribution are not random, but show a systematic tendency of increases in Ne0 as D decreases” does not necessarily support the accuracy or validity of the model.

3. The small values of the model-fitted Ne0 are used to conclude the existence of ionosphere “holes”. To assure the fitted Ne0 values are reasonable, it would be beneficial to correlate the Ne0

with the cutoff frequency of the JDPs. If a positive correlation exists between the two, the inferred results would be more believable, at least qualitatively. The authors have the information of the cutoff frequencies and should have used them.

4. The authors claim two peaks in figure 2b for the inter-pulse time separation. A statistical significance analysis is needed to support the claim.

Minor comments:

1. Line 53. Specify the frequency range here. I found it to be 150 kHz much later in the text.

2. Line 57. Suggest change "All of the pulses were detected at altitude between 9790 km and 316,000 km above the 1 bar level." to "All of the pulses were detected while Juno was at altitude between 9790 km and 316,000 km above the 1 bar level". I first thought the radio sources were at these altitudes.

3. Lines 57-59, sentence starts with "These...". Why is this important? The radio propagation would be affected by the entire path from the source to Juno but not only the local condition at Juno, unless the entire path can be assumed to have the same f_{ce} and f_{pe} .

We appreciate the reviewers for a careful reading of the manuscript and helpful comments and suggestions. Taking your comments and suggestions into consideration, we have answered your questions in this report and revised our manuscript accordingly. We dropped all in-text citations and restricted the abstract to 150 words, according to *Nature Communications* guidelines. Our changes can be tracked with underlined and highlighted text in the revised manuscript.

Reviewers' comments:

Reviewer #1 (Remarks to the Author):

The dispersed pulses in Jovian atmosphere should prove to be of big interest to the scientific community, especially if as the authors suggest they are propagating through the ionospheric density holes.

In short, the paper is well-written and structured and I recommend publication after minor fixes.

Below follows a list of comments and suggestions, in case the attached PDF fails.

L51: A short introduction is necessary for most readers, describing why JDPs are interesting and what they can reveal

We added two introductory paragraphs in the revised manuscript.

L56: "Jovian radii" - add $1R_J = \dots$ for convenience?

We added "(R_J , $1 R_J = 71492$ km)."

L66: perhaps a short recap of the modes?

Done.

L71: $f_{pe} = 40$ kHz - what is the basis of this assumption?

We do not know the exact plasma frequency, but the basis of this assumption comes from the highest frequency of the smooth emission at 40 kHz below the JDP which we have determined to be the local plasma frequency. In order to clarify this point, we modified the text as follows:

"Another observation is that JDPs can be found above the maximum Z-mode frequency, the upper hybrid frequency $f_{uh} = \sqrt{f_{pe}^2 + f_{ce}^2}$; for example, the maximum frequency of the JDP in Fig. 1c is at least 150 kHz, well above both $f_{ce} = 126$ kHz and $f_{uh} = 132$ kHz (assuming $f_{pe} = 40$ kHz as the highest intensity of the smooth emission below the JDP)."

L73: phenomenology = nature?

Done.

L78: what about the other 47%? (do they have more than 16 ms between? why not include them too?)

The other 47% of detections show only one pulse in a waveform snapshot. The snapshots have a duration of 16.384 ms and are obtained once per second, hence, we cannot measure inter-pulse spacings between 16 ms and one second. We modified this sentence as: “It utilizes only snapshots containing two or more pulses allowing inter-pulse spacing. Hence, only 53% of the detections (236 counts) are included.”

L80: which original set?

The original set is all detected pulses, 445. We modified “original set” as “complete set”.

L86-88: from the figure it seems that there are actually two distributions, this is very interesting and should be discussed

Given the uncertainty of the model fitting (Supplementary Fig. 4), it may not be reasonable to discuss the detailed distributions in Fig. 2c. Rather, we briefly mentioned the general trend of the distribution in these sentences.

L97: this is very specific, why 300km?

We use 300 km altitude as the lowest Jovian ionosphere measured by Voyager 2 and Galileo (Hinson et al., 1997, 1998). We added this point as follows: “Assuming that whistlers propagate parallel to the magnetic field (so-called “ducted” whistlers)³, the footprints of these whistlers can be estimated by mapping from Juno’s position along the modeled JRM09 magnetic field lines¹⁹ onto the Jovian atmosphere at 300 km altitude (the bottom edge of the Jovian ionosphere measured by Voyager 2²⁰ and Galileo²¹) above the 1-bar level.”

L108-111: this is a bias then, can it be corrected for? perhaps by filtering the longitudes with high auroral emissions (and then showing in an extra figure)?

We re-investigated the possibility that the high signal levels from auroral radio emissions affect no detections of JDPs from 0° to 210° in the southern hemisphere. In fact, there were the low number of JDP detections when Juno's near-source crossings typically occurred at $\sim 5\text{-}6 R_J$ and most of the detections occur within $\sim 5 R_J$. Therefore, the auroral radio observations do not directly cause the absence of JDPs at this region. We now considered the possibility that a lack of the LFR-Hi observations for PJ1 and PJ3 causes the above issue. In order to remove this observation bias, we selected all data only when the LFR-Hi observations were available and re-organised the distributions of JDPs, whistlers, and sferics in Supplementary Fig. 7. As a result, we still detected the whistlers and sferics at this region. Hence, the lack of the LFR-Hi observation coverage has nothing to do with the absence of JDP detections in this region. Another possibility is a lack of ionospheric holes or dense ionosphere at this region because the JDPs can be seen only when there is a low density path through the ionosphere that allows these pulses to reach Juno's position. Further consideration is beyond the scope of this paper. We may be able to examine this in the future by including more data. Accordingly, we modified these sentences as follows: "The longitude range of 0° to 210° in the southern hemisphere shows no JDPs, contrary to the whistler and sferic distributions. Even if we consider the limited LFR-Hi observation coverage, this trend remains (Supplementary Fig. 7). Hence, the lack of JDPs could be due to a dense ionosphere between the JDP radio source and Juno."

L112: bad start of a sentence, consider rephrasing, e.g., "the locations of the JDP source are... suggesting..." or similar

We modified the sentence as follows:

"In addition to two concurrent JDP-sferic events (Supplementary Fig. 8), the JDP source locations are similar to those of lightning-induced whistlers and 600-MHz sferics, suggesting JDPs are related to lightning."

L128-132: Question: Is it a "trivial" knowledge that the lightnings occur all the time, but the pulses are seen only when an ionospheric "hole" is present allowing the wave to pass? If this is the case, I think it would be useful to highlight this somewhere here - using JDPs as hole detectors should prove useful.

We added a sentence as follows:

"Recall that, while the 600-MHz sferics propagate freely through the dense ionosphere, the JDPs can be seen only when there is a low density path through the ionosphere that allows these pulses to reach Juno's position."

Figure 2: consider recoloring the cum. prob. axes to orange for consistency

We changed the colour of the cumulative probability axes into orange in Figs. 2a and 2b.

Figure 2: considering that there are no counts above 13 ms, the histograms could be "zoomed" a bit to make the peaks more distinguishable - unless 16 ms is somehow important?

We added the following sentence:

“Note that the maximum of 16 ms for both duration and inter-pulse spacing is due to the length of the Waves LFR-Hi waveform snapshot.”

L182-183: errorbars in the figure to match these would make interpretation much easier, even if it's just one generic "legend" of uncertainty (i.e., for one point)

We updated Figs. 2c and 2d by including each error of one standard deviation from the model fittings. Also, we added the results of the error analysis of N_{e0} , D , and C in Supplementary Fig. 4. Accordingly, we modified this sentence as follows:

“The grey error bar corresponds to one standard deviation (68% confidence interval) of N_{e0} and D . The median values of the one standard deviation from the model fittings are $27.5\% \pm 2.5\%$ for D and $2.5\% \pm 2.5\%$ for N_{e0} (Supplementary Fig. 4).”

Figure 3: the axis on the right plot is a bit too far from the labeled y-axis, consider adding tick values here as well

We added tick values and label in the ordinate of this plot.

Supplementary Figure 1: with this orientation of the figure, all of the y-axes appear upside down, which is a bit inconvenient to read - consider rotating the figure clockwise, or rotating the y-axes labels

We fixed this figure by rotating clockwise.

Supplementary Figure 2: the blue bars seems superfluous (and confusing), if the spheric events are detected all the time during these intervals, why not just indicate this?

We deleted the blue bars in this figure.

Reviewer #2 (Remarks to the Author):

Review for “Evidence for low density holes in Jupiter’s ionosphere” by Imai et al.

This paper presents the first detection of a new form of radio emission, Jupiter dispersed pulses. The authors hypothesise that these radio pulses are linked to jovian lightning bursts due to the coincidence of events with sferics and Whistler waves.

The paper is interesting - new radio emission forms are quite exciting! - and worthy of publication after the following points are addressed:

Line 78: Figure 2b contains only 53% of the detections as it focuses on repeating JDP pulses with a repetition time less than 16 ms. What about the remaining 47% of the data? Are those events single pulses? What is the importance of this inter-pulse spacing? Does it strengthen the case for lightning bursts driving the emission?

The other 47% of the detections are where a single pulse appears in a 16-ms waveform snapshot. Therefore, we cannot measure inter-pulse spacing larger than 16 ms. We explicitly added this caveat in this sentence and caption of Fig. 2. The important implication of the inter-pulse spacing is discussed in the last paragraph of the main text in terms of a similarity of JDPs with the trans-ionospheric pulse pairs (TIPPs) at Earth.

Line 106 - 111: What about the region between 200 to 280 degrees where JDPs are present in the southern hemisphere, but neither sferics nor Whistlers are observed? There are additional locations where there are unique events of each type. It’s not obvious that the correlation between JDPs and sferics is as strong as suggested by the authors.

This region corresponds to the samples from the Juno PJ9 orbit and we only use it for the JDP surveys because, during this pass, the JDPs first appeared above the local electron cyclotron frequency (e.g. Fig. 1c). For the other two emissions, we use the data on PJ1 and PJ3 to PJ8 through the published catalogues produced by Kolmašová et al. (2018) and Brown et al. (2018). We added this comment in the caption of Fig. 3:

“Note that the region between 200° and 280° where JDPs present was sampled from the Juno perijove 9 orbit where sferic and whistler observations have not been completed.”

More importantly, even though the three phenomena are commonly lightning-induced electromagnetic waves, they propagate differently to the observer. The whistlers may propagate away from lightning strokes below the ionosphere before becoming ducted. The 600-MHz sferics and JDPs come from the strokes but, while the 600-MHz sferics propagate freely through the dense ionosphere, the JDPs can be seen only when the ionospheric density is low enough to allow them to propagate through the ionosphere. Hence, in this study, we conclude that the occasional concurrent observations of these is evidence for a common source in lightning.

Line 130 - 132: Is the lack of JDPs observed around the jovicentric equator an observational artefact due to the upper recordable frequency of the instrument? This upper limit has already

been mentioned in line 68 as the reason that only 2% of cases are seen with frequencies above fce. Clarification is needed.

The reason for no JDP detections around Jovicentric equator is because the ionospheric density around the equator is too dense to allow JDPs below 150 kHz propagate into the magnetosphere. We clarified this point in the sentences, below:

“No JDPs are observed around the Jovicentric equator because the local plasma frequency of the Jovian ionosphere tends to exceed the upper recordable frequency of 150 kHz at latitudes from -10° to 25° and the JDP L-O mode waves cannot propagate in this region.”

Supplementary figure 3 and associated discussion in lines 141 - 143: Due to Juno’s orbit, measurements of JDPs and sferics is biased towards the terminator. The discussion should be tempered with respect to the local time coverage or that bias should be more explicitly stated in text. The phrasing “it is possible that the radio locations of JDPs observed by Juno and postulated ionospheric holes are both on the night side where a different recombination process from the day side is anticipated” is unclear.

We modified these sentences as follows:

“Juno’s first nine orbits used in this study were concentrated near the terminator (Supplementary Fig. 9). It is possible that the actual radio locations of JDPs observed by Juno and postulated ionospheric holes are on the night side, where a different recombination process from the day side is anticipated²³.”

Reviewer #3 (Remarks to the Author):

Comments on “Evidence for low density holes in Jupiter’s ionosphere” by Imai et al

This reviewer found that the paper could be interesting to multidisciplinary readers but the work lacks strong and definitive evidence and analysis to support the conclusions.

Major comments:

1. The authors assumed the sources for the JDPs were produced by lightning below Juno, which might be correct but needs more definitive data to unambiguously support their claim. In figure 3 the authors compared the Juno positions when the JDPs were detected with the whistler and MWR (500 MHz) source positions and claimed that the JDPs were detected in the same general regions and inferred them as related to lightning. It is not clear from the main text and the supplementary material if the three types of observation were coordinated in time and space. Coordinated or not, the majority of the JDPs appear to be detected along different Juno orbital trajectories as compared to the other two observations. If they were coordinated, one should expect a MWR detection for each of the JDPs. The authors provided two events in Supplementary figure 2 that appear to be detected at the same time (within 100 ms) and position by MWR. However, 2 out of 445 appears to be a very small fraction and the authors need to rule out the chance of random coincidence.

We cannot determine the exact position of JDP sources or the direction of arrival of JDP with one electric component measured from Waves. We simply assume that the JDP sources were directly below Juno when these pulses were detected. As a consequence, the determined JDP source locations follow the Juno orbits. Fig. 3 shows the statistical distribution of JDPs, whistlers, and sferics because Waves and MWR operate independently and, by chance, these instruments sometimes performed coincident observations of lightning. However, the Waves LFR-Hi band for JDPs and LFR-Lo band for whistlers cannot be simultaneously recorded due to the instrumental design (Kurth et al., 2017). Therefore, either coincident observations of whistlers and sferics (Imai et al., 2018) or coincident observations of JDPs and sferics (Supplementary Fig. 8) are possible. Furthermore, we found 39 MWR sferic 100-ms events that overlap with Waves 16.384-ms snapshots but there were only two Waves snapshots in which we detected JDPs. These concurrent events are shown in Supplementary Fig. 8. One reason for a small number of concurrent event is a different propagation scenario. While the 600-MHz sferics propagate freely through the dense ionosphere, the JDPs can be seen only when ionospheric holes are present. Another possibility is an unclear number of sferics during a Waves snapshot. While Waves can clearly capture individual JDPs in a 16.384-ms waveform snapshot, there is no way to identify exactly when individual sferics occur within the MWR sferic 100-ms integration time. These observational and instrumental restrictions most probably limit the number of the concurrent JDP and sferic events. We added these comments in the caption of Supplementary Fig. 8.

2. The dispersion model (assuming it good enough for the ionosphere) used by the authors in Method is more sensitive to D than to Ne0 to fit the observed dispersion curve, therefore, one would expect D to have a smaller scattering range than Ne0. However, figure 2c shows the

opposite with D scattering in ~ 4 orders of magnitude and N_{e0} in ~ 2 orders of magnitude. In addition, the paper lacks the necessary error analysis for D , N_{e0} and C , making the critical results of N_{e0} less confident. If a Taylor expansion is used to approximate the authors' model, one can see that the dispersion is proportion to the product of D and N_{e0} to the first order, and therefore will naturally leads to the general feature of figure 2c (regardless of the scattering ranges), i.e., a smaller D will require a larger N_{e0} . So the authors' statement in line 86 "The distribution are not random, but show a systematic tendency of increases in N_{e0} as D decreases" does not necessarily support the accuracy or validity of the model.

If we repletely observed the similar spectral shapes of JDPs, D would have been more sensitive than N_{e0} . However, as Fig. 1 shows various types of JDPs, this simple expectation is not appropriate. We made the correlation analysis for N_{e0} , D , and C in Supplementary Fig. 5 and found strong negative correlations for the pairs of N_{e0} and D and of D and C and strong positive correlation for the pair of C and N_{e0} . Therefore, we confirmed that the tendency of increasing in N_{e0} as D decreases is due to fittings of the O mode propagation model. We also added Figs. 2c and 2d with one standard deviation (68% confidence interval) of D and N_{e0} to fit the O mode propagation model, and included the histograms of fractions of the model parameters in Supplementary Fig. 4. Nevertheless, the O mode propagation model allows to express various types of spectral structures. Supplementary Fig. 6 shows eight examples of simulated dispersed pulses from the O mode propagation model. Each colour dispersed pulse in Supplementary Fig. 6a correspond to each set of D and N_{e0} in Supplementary Fig. 6b. The nature of JDPs appears the limited sets of D and N_{e0} (e.g. the orange, blue, pink, and sky blue dispersed pulses). In contrast, the green, red, purple, and brown dispersed pulses were not detected in our study. Taking into account the aforementioned arguments, we modified the sentences in the main texts as follows. "The distributions show a systematic tendency of increases in N_{e0} as D decreases due to the strong negative correlation of determinations of both parameters in the model (Supplementary Fig. 5). Nevertheless, these distributions provide possible solutions to characterise the JDP spectral shapes (Supplementary Fig. 6)."

Also, we added the following sentence in the end of Methods:

"Eight examples of simulated dispersed pulses are displayed in Supplementary Fig. 6."

3. The small values of the model-fitted N_{e0} are used to conclude the existence of ionosphere "holes". To assure the fitted N_{e0} values are reasonable, it would be beneficial to correlate the N_{e0} with the cutoff frequency of the JDPs. If a positive correlation exists between the two, the inferred results would be more believable, at least qualitatively. The authors have the information of the cutoff frequencies and should have used them.

We added Supplementary Fig. 3, which shows the distributions of f_{pe0} estimated from the O mode propagation model and f_{cutoff} measured from the spectrograms. We found that the Pearson correlation coefficient r is 0.89, a positive correlation. We added the following sentence in the main text:

"Additional support for the fit results comes from a positive correlation between f_{pe0}

estimated from the model and f_{cutoff} measured from the spectrograms (Supplementary Fig. 3).”

4. The authors claim two peaks in figure 2b for the inter-pulse time separation. A statistical significance analysis is needed to support the claim.

We attempted to fit the inter-pulse spacing distributions with one modified log-normal distribution and two modified log-normal distributions. The results are shown in Supplementary Fig. 2. The coefficient of determination is higher for two distributions than for one distributions. Hence, each distribution has one peak and there are two peaks in the inter-pulse spacing histogram.

Minor comments:

1. Line 53. Specify the frequency range here. I found it to be 150 kHz much later in the text.

We modified the sentence as follows:

“We have carried out a survey of the Juno Waves burst mode data from the Low Frequency Receiver High (LFR-Hi) channel in the form of frequency-time spectrograms below 150 kHz (Methods) on PJ1 and PJ3 to PJ9.”

2. Line 57. Suggest change “All of the pulses were detected at altitude between 9790 km and 316,000 km above the 1 bar level.” to “All of the pulses were detected while Juno was at altitude between 9790 km and 316,000 km above the 1 bar level”. I first thought the radio sources were at these altitudes.

We changed the recommended sentence.

3. Lines 57-59, sentence starts with “These...”. Why is this important? The radio propagation would be affected by the entire path from the source to Juno but not only the local condition at Juno, unless the entire path can be assumed to have the same f_{ce} and f_{pe} .

We deleted this sentence.

REVIEWERS' COMMENTS:

Reviewer #1 (Remarks to the Author):

No further comments, I am satisfied with the authors' response and editing.

Reviewer #2 (Remarks to the Author):

Thank you for addressing the concerns of myself and the other referees so thoroughly. The paper now clearly reports the first observations of JDPs to the community and is worthy of publication.

Reviewer #3 (Remarks to the Author):

The authors have addressed all my comments properly through additional analysis. I recommend it publication in its current version.

REVIEWERS' COMMENTS:

Reviewer #1 (Remarks to the Author):

No further comments, I am satisfied with the authors' response and editing.

Reviewer #2 (Remarks to the Author):

Thank you for addressing the concerns of myself and the other referees so thoroughly. The paper now clearly reports the first observations of JDPs to the community and is worthy of publication.

Reviewer #3 (Remarks to the Author):

The authors have addressed all my comments properly through additional analysis. I recommend it publication in its current version.

We appreciated thoughtful comments of the three reviewers.